# Modeling and Verification of Rolling Resistance Torque of High-Speed Rubber Track Assembly Considering Hysteresis Loss

**DOI:** 10.3390/polym15071642

**Published:** 2023-03-25

**Authors:** Kang Liang, Qunzhang Tu, Xinmin Shen, Juying Dai, Qin Yin, Jinhong Xue, Xuan Ding

**Affiliations:** 1Field Engineering College, Army Engineering University of PLA, Nanjing 210007, China; 2State Key Laboratory of Intelligent Manufacturing of Advanced Construction Machinery, Xuzhou Construction Machinery Group, Xuzhou 221004, China

**Keywords:** rubber track, rolling resistance, thermo-mechanical, viscoelastic, energy consumption

## Abstract

Due to the viscoelasticity of rubber materials, hysteresis loss due to deformation is the main reason for the rolling resistance of high-speed rubber tracks. Since the structure and material of high-speed rubber track assemblies are different from traditional tires and metal tracks, the rolling resistance theory of traditional wheeled and tracked vehicles is not applicable. Therefore, in order to determine the rolling resistance scientifically and accurately, the mechanism research of the rolling resistance of high-speed rubber track assembly is the key to the design of high-speed rubber crawler vehicles. In this paper, the stress–strain characteristics of rubber track under the action of compression, tension, bending, and driving were studied. The strain load spectrum of rubber tracks was established, and the strain cyclic load was extracted by the rainflow method. The temperature model of the rubber track was developed based on its dynamic characteristics. On the basis of energy conservation, the hysteresis loss of rubber is equivalent to the energy consumption of rolling resistance, and the theoretical model of rolling resistance of high-speed rubber track assembly is established. In accordance with the model above, the key influencing factors and changing trends of rolling resistance are analyzed, which provides a theoretical basis for the performance optimization of high-speed rubber track assembly.

## 1. Introduction

With the gradual strengthening of environmental awareness and the increasing risk of energy crisis, energy conservation has become the unremitting pursuit of global researchers. Vehicle rolling resistance directly determines the rationality of power matching and fuel consumption, which is the key research object and improvement index of vehicle-related researchers [1,2]. Because the structural form is different from traditional tires and the material is different from metal tracks, the rolling resistance theory of traditional wheeled vehicles and metal tracked vehicles is not applicable to the study of rolling resistance of high-speed rubber track assembly. There is no mature theory on the rolling resistance mechanism and change law of high-speed rubber track assembly.

Aiming at the rolling resistance of low-speed rubber track assembly, Chołodowski et al. pointed out that the rolling resistance of rubber track assembly is mainly composed of bending resistance, guiding resistance, load wheel settling resistance, and bearing friction resistance. The rubber track assembly test bench was built to test the influence of factors such as wheel train diameter, load wheel suspension, tension force, and load on rolling resistance. Based on the experimental data, the empirical formula of rolling resistance fitting was given. The analysis shows that the energy loss caused by the first-order modal vibration of the rubber track accounts for only a small part of the total energy consumption of the rubber track walking system [3,4,5,6,7]. Using a special test bench for rubber track assembly, Based on the test data, the above study gives the influence of track train and tension on the trend of rolling resistance. However, the mechanism of rolling resistance of high-speed rubber track assembly has not been studied. The existing research results could not support the performance optimization of high-speed rubber track assembly.

In view of the rapid heating of rubber under alternating load, domestic and foreign researchers have carried out a large number of studies on the temperature rise mechanism of rubber hysteresis. Medalia pointed out that at the macro level, hysteresis loss is a specific manifestation of the viscoelastic characteristics of rubber materials. The deformation of rubber materials can be divided into elastic energy storage and viscous loss. When the rubber is subjected to alternating loads, the viscous loss part turns into heat, causing the temperature of the material to rise. At the microscopic level, viscoelasticity essentially stems from the internal friction behavior of the molecular chains inside the material and the microstructure of the filler lamp [8]. Akutagawa et al. proposed the mechanism and theoretical model of internal energy loss of rubber due to hysteresis characteristics under alternating loads. When a load is applied to the rubber specimen and causes the rubber to deform, the work is stored inside the rubber in the form of energy, including reversible and irreversible processes. The reversible thermal process that causes the temperature increase is partially described by the configuration entropy process. The irreversible process is partially transferred to the fluctuations in the movement of molecules inside the rubber, manifested by stress relaxation [9]. Garnier et al. studied the effects of cyclic loading and coupling heat on the viscoelastic response of carbon-black-filled nitrile rubber. Cyclic loading tests of different thermomechanical loads at different temperatures were carried out to analyze the relative contribution of mechanical and thermal loads and their coupling to the viscoelastic response. The result shows that the modulus of energy storage increases with the increase number of cycles due to the increase crosslinking. At the same time, the cyclic mechanical load will significantly enhance the influence of temperature on the mechanical properties of rubber materials and form a thermodynamic coupling [10]. In order to overcome the problem that complex local mechanical structures are difficult to predict heat sources and temperature fields, Le Saux et al. proposed a simple phenomenological method. By a single test of an hourglass-shaped specimen, the thermal and mechanical parameters can be effectively identified, and the influence of displacement amplitude, frequency, load ratio, specimen volume, and other factors in the tensile and torsional test of the hourglass-shaped specimen can be analyzed [11].

Tires and rubber tracks are rubber composite materials composed of rubber matrix and cord reinforcement, which are subject to alternating loads during driving. Due to the similar structure and bearing characteristics of tires and rubber tracks, the theory of tire rolling resistance can be used as a reference for the study of the rolling resistance mechanism of rubber track assembly. Based on the nonlinear viscoelastic theory and the thermal coupling finite element numerical analysis method, Li et al. studied the static three-dimensional contact problem of tire pavement. The two-dimensional axisymmetric model was used to study the transient temperature rise behavior and rolling resistance characteristics of solid rubber tires and the functional relationship between heat source and loading strain, frequency, temperature, and loading times. Combined with the relationship between rubber thermal parameters and temperature, it is used to predict tire rolling resistance and transient temperature distribution. The quantitative effects of thermal conductivity and loss factors on tire temperature rise and rolling resistance under different loading displacement and rotational speed conditions are simulated [12,13]. For ME-Wheel mechanical elastic wheels, Zhu et al. carried out a force balance state analysis. Combined with the dynamic characteristics such as the loss factor of rubber materials, the thermal balance analysis was carried out, and the theoretical model of heat generation and heat dissipation of wheel edge rubber was established. The thermal equilibrium temperature is calculated according to the above model, and the effectiveness of the model is verified by experiments [14]. For the new non-pneumatic tire with a flexible spoke structure, Fu et al. used the unit configuration method to construct a 3D model of the flexible spoke non-pneumatic tire. The thermal-continuous coupling method was used to analyze the deformation, energy loss, and heat conduction of flexible spoke non-pneumatic tires. The steady-state temperature field distribution under heat-engine coupling is obtained. The result shows that the high temperature zone is mainly distributed in the middle bending part of the flexible spoke unit of non-pneumatic tires, and the temperature gradually decreases from the center to the periphery. The influence of load on the temperature change of non-pneumatic tires is greater than the driving speed, which provides a theoretical basis and method guidance for solving the failure problem of non-pneumatic tires under thermodynamic coupling [15,16]. Many scholars have developed the influence of temperature, road conditions, pattern types, and other factors on the rolling resistance of tires. The research method and change trend of rolling resistance of rubber tires are given, which provides a basis for the research of rolling resistance of rubber tracks [17,18,19,20,21,22,23,24,25]. The above studies show that rubber tracks are subjected to high-frequency alternating loads during driving. Due to the temperature sensitivity of rubber materials, the thermodynamic coupling effect of rubber tracks cannot be ignored. The influencing factors of the rolling resistance of the rubber track assembly are studied, and the test data fitting formula is given. However, there is a lack of research and theoretical model on the rolling resistance mechanism of high-speed rubber track assembly, which cannot support the determination and optimization of rolling resistance of high-speed rubber track assembly.

In this paper, the load spectrum of rubber track during driving is established, and thermal coupling analysis is carried out. A new theoretical model of the rolling resistance of the rubber track assembly considering the hysteresis energy loss is proposed. In this model, the lag energy loss of the rubber track is innovatively equivalent to the energy consumption of the rolling resistance of the track assembly, providing a new method for the study of the rolling resistance of the high-speed rubber track assembly. The influencing factors and change laws of high-speed rubber track assembly are given to support the performance optimization of high-speed rubber track assembly.

## 2. Model Development

### 2.1. Research Protocol

Based on the viscoelastic theory, a heat source is formed inside the rubber under alternating loads. At the same time, the increase temperature will affect the mechanical properties of the rubber material and then change the internal stress balance state of the rubber. Taking the test data of wheel train structure size, load, driving speed, and rubber dynamic mechanical properties as the initial parameters and considering the hysteresis energy loss, the rolling resistance research scheme of high-speed rubber track assembly was formulated.

As shown in Figure 1, based on the structure and load parameters of the high-speed rubber track assembly, the force analysis of the drive wheel, tensioner, load wheel, and idler train is carried out. Rubber track strain models such as drive, collapse, bending, and tensioning are established. The cyclic operation process of rubber tracks was analyzed, and then the strain load spectrum of rubber tracks was established. The rainflow method is used to extract the cycle amplitude of the strain load, and the strain rainflow matrix during driving is established. Combined with the dynamic mechanical properties of rubber materials, the dynamic heat generation model of rubber tracks was established. Considering the heat dissipation process of the rubber track during driving, the heat dissipation model of the rubber track was established. Combined with the heat generation model and the heat dissipation model, the comprehensive temperature rise law of the rubber track was given. The rising track temperature will further affect the dynamic mechanical properties of the material. The hysteresis heat generation of the rubber track is equivalent to the rolling resistance energy consumption. Combined with hysteresis equivalent rolling resistance, bearing friction resistance, inertial resistance, and wind resistance, the rolling resistance model of high-speed rubber track assembly was established.

### 2.2. Load Analysis of High-Speed Rubber Track Assembly

As shown in Figure 2, the high-speed rubber track assembly studied in this paper mainly includes drive wheels, tension springs, tension arms, tension wheels, load wheel set 1, load wheel set 2, and rubber tracks. The drive wheels are located in the upper part of the high-speed rubber track assembly. The drive wheels are connected to the frame by means of bearings. The drive wheels engage with the track drive teeth through drive pins to transmit the driving force. The tensioner is located at the bottom of the triangle, and the rubber tracks are tensioned by tension arms and tension springs. Load wheel set 1 and load wheel set 2 are located at the bottom of the track assembly. The weight of the vehicle is transmitted to the rubber tracks via load wheel set 1 and load wheel set 2. The load wheel set 2 contains a load wheel 3, a load wheel 4, and a balance beam 2. The load wheel set 2 is articulated with the hinge point 2 of the rack through articulation hole 2. The balance beam 2 drives the load wheel 2 to rotate around hinge point 2. The load wheel set 2 contains load wheel 3, load wheel 4, and the balance beam 2. The load wheel set 2 is articulated with the hinge point 2 of the rack through articulation hole 2. The balance beam 2 drives the load wheel set 2 to rotate around hinge point 2. The load wheel4 also serves as a guide, limiting lateral displacement of the tracks and avoiding dechaining of the rubber tracks.

The bearing weight $G_{0}$ of the axle and the weight $G_{1}$ of the assembly itself are the main loads of the high-speed rubber track assembly. The bearing weight of the axle is transmitted to the ground by means of drive wheels, drive wheel bearings, frames, load wheel bearings, load wheels, and rubber tracks. The high-speed rubber track assembly’s own weight $G_{1}$ is also transmitted to the ground via a load wheel. The load of the high-speed rubber track assembly is transferred to the rubber track using a two-stage load distribution. The first stage load distribution consists of hinge point 1, hinge point 2, and the frame. The first stage load distribution enables load distribution from the high-speed rubber track assembly to hinge points 1 and 2. Therefore, according to the bearing and structural parameters of the high-speed rubber track assembly, the load of the high-speed rubber track assembly load wheel can be expressed as shown in Equation (1) [26].
(1)$$\left[\begin{array}{c} F_{1}\\ F_{2}\\ F_{3}\\ F_{4} \end{array}\right]=\left[\begin{array}{cc} 1-\beta_{1} & 0\\ \beta_{1} & 0\\ 0 & 1-\beta_{2}\\ 0 & \beta_{2} \end{array}\right]\left[\begin{array}{cc} 1-\alpha_{1} & 1-\alpha_{2}\\ \alpha_{2} & \alpha_{1} \end{array}\right]\left[\begin{array}{cc} G_{0} & 0\\ 0 & G_{1} \end{array}\right]\left[\begin{array}{c} 1\\ 1 \end{array}\right]$$
where, $G_{0}$ is axle load; $G_{1}$ is the weight of the high-speed rubber track assembly; $G_{2}$ is the load of hinge point 1; $G_{3}$ is the load of hinge point 2; $F_{1}$,${\;F}_{2}$,${\;F}_{3}$ and $F_{4}$ is the load of load wheel 1–4 respectively; $L_{1}$ is the distance between $G_{0}$ and $G_{2}$ along the X axis; $L_{2}$ is the distance between $G_{0}$ and $G_{3}$ along the X axis; $L_{3}$ is the distance between $G_{1}$ and $T_{1}$ along the X axis; $L_{4}$ is the distance between $G_{1}$ and $T_{2}$ along the X axis; $L_{11}$ is the distance between $G_{2}$ and $F_{1}$ along the X axis; $L_{12}$ is the distance between $G_{2}$ and $F_{2}$ along the X axis; $L_{21}$ is the distance between $G_{3}$ and $F_{3}$ along the X axis; $L_{22}$ is the distance between $G_{3}$ and $F_{4}$ along the X axis; $\alpha_{1}$ is the ratio of $L_{1}$ to $\left(L_{1}+L_{2}\right)$; $\alpha_{2}$ is the ratio of $L_{3}$ to $\left(L_{3}+L_{4}\right)$; $\beta_{1}$ is the ratio of $L_{11}$ to $\left(L_{11}+L_{12}\right)$; $\beta_{2}$ is the ratio of $L_{21}$ to $\left(L_{21}+L_{22}\right)$; $\alpha_{1}$ and $\alpha_{2}$ values are between 0 and 1; $\beta_{1}$ and $\beta_{2}$ values are between 0 and 1.

### 2.3. Establish the Constitutive Model of Rubber Track

According to the theory of composite laminates, composite materials are defined as multilayer materials. In the small deformation state, the overall material performance is estimated by the superposition method between them.

As shown in Figure 3, the rubber track is composed of steel wire rope and rubber matrix, which is applicable to the composite laminate theory. The material of the rubber matrix is isotropic. The material properties of wire rope are anisotropic. The constitutive equation of rubber track is the comprehensive superposition of the constitutive equation of wire rope core and the constitutive equation of rubber matrix.

#### 2.3.1. Normal Constitutive Model of Rubber Track

When the rubber track is normally loaded, the wire rope core is laterally loaded, and the influence of the wire rope core on the normal constitutive model of the rubber track can be ignored [27]. Therefore, the normal constitutive model of the rubber track can be simplified to the constitutive model of rubber matrix, as shown in Equation (2).
(2)$$E_{\mathrm{rubber}}\left(w,T\right)=\sqrt{E^{\prime }{\left(w,T\right)}^{2}+E^{\prime \prime }{\left(w,T\right)}^{2}}$$

According to the WLF (Williams–Landel–Ferry) function [28], the influence of temperature T on the elastic modulus $E\left(f,T\right)$ of the rubber can be transformed into the influence of the disturbance frequency f, as shown in Equations (3) and (4). Based on the temperature spectrum of rubber material obtained from the test, the dynamic mechanical properties of rubber under the influence of temperature and frequency can be obtained.
(3)$$E\left(f,T\right)=E\left({f\alpha }_{T}/\alpha_{T_{0}},T_{0}\right)$$
where
(4)$$\left\{\begin{array}{c} \log_{10}a_{T}\approx -8.86\frac{T-T_{s}-50}{51.5+T-T_{s}}\\ \log_{10}a_{0}\approx -8.86\frac{T_{0}-T_{s}-50}{51.5+T_{0}-T_{s}} \end{array}\right.$$
where $\alpha_{T}$ is the shift factor at temperature T; $\alpha_{0}$ is the shift factor at temperature $T_{0}$; $T_{s}$ is the reference temperature of the rubber material.

#### 2.3.2. Circumferential Constitutive Model of Rubber Track

When the rubber track is subjected to circumferential overall tension, the tensile deformation and strain of rubber and steel fiber are the same. Therefore, the circumferential load of rubber track can be expressed as Equation (5).
(5)$$F_{S}=E_{S}A\varepsilon =\left(E_{\mathrm{wire}}A_{\mathrm{wire}}+E_{\mathrm{rubber}}\left(w,T\right)A_{\mathrm{rubber}}\right)\varepsilon$$

According to GB/T 20786-2015, the tensile strength of rubber track wire rope shall not be less than 5 times the weight of the whole vehicle. The steel wire rope in the rubber track is elastic deformation during driving. Therefore, wire rope is regarded as an elastic material.

As shown in Equation (6), the equivalent elastic modulus of the rubber track can be expressed as:(6)$$E_{S}=\frac{E_{\mathrm{wire}}A_{\mathrm{wire}}+E_{\mathrm{rubber}}\left(w,T\right)A_{\mathrm{rubber}}}{A}$$
where $A_{\mathrm{wire}}$ is the cross-sectional area of the wire rope; A is the cross-sectional area of rubber track; $A_{\mathrm{rubber}}$ is the cross-sectional area of the rubber substrate; $E_{\mathrm{rubber}}$ is the elastic modulus of rubber matrix; $E_{S}$ is the elastic modulus of rubber track; $F_{S}$ is the circumferential tension of rubber track; $E_{\mathrm{wire}}$ is the elastic modulus of wire rope core, with the value of 12,000 Mpa [29]; $E_{\mathrm{rubber}}\left(w,T\right)$ is the elastic modulus of rubber, which is obtained from the test.

### 2.4. Establish the Load Spectrum

The load spectrum of rubber track is used to describe the change trend of rubber track load with time during driving. The load spectrum is related to many factors, such as load, stress position, gear train diameter, tension, and track material. The load of the rubber track changes in cycles during driving. Due to the viscoelasticity of rubber, the hysteresis loss in the process of cyclic loading will be converted into heat. In this section, the driving strain, tension strain, bending strain, and compression strain of the rubber track during driving are analyzed. The track load spectrum is constructed to provide the basis for the subsequent rolling resistance research.

#### 2.4.1. Indentation Strain Analysis

The road wheel is the wheel body that bears the weight of the high-speed rubber track assembly and the vehicle and is the largest load bearing of all wheel bodies. The indentation strain is the strain produced by the compression of the inner surface of the road wheel and the rubber track. During the rolling contact between the road wheel and the rubber track, the material located in the forward direction of the contact area is compressed, and the material leaving the area will relax and recover. Depending on the viscoelasticity of the rubber material, there is a hysteresis in the elastic recovery of the material, resulting in a hysteresis energy loss.

As shown in Equation (7), based on the Winkler elastic theory model, the settlement of any point on the contact surface is proportional to the pressure per unit area of the point. Any position within the contact area is reduced to an independent spring on a rigid base. Settlement occurs only in the area under pressure, and no settlement occurs in other unstressed areas.
(7)$$P\left(x,y\right)=k\cdot w\left(x,y\right)$$
where $P\left(x,y\right)$ is the pressure of the contact area; $w\left(x,y\right)$ is the subsidence of the contact area; k is the stiffness coefficient.

Due to the viscoelasticity of rubber materials, the dynamic mechanical properties are greatly affected by the excitation frequency and temperature, and the traditional Winkler elastic theory model cannot meet the analysis requirements. Therefore, as shown in Figure 4, based on the dynamic mechanical properties of rubber materials, an improved Winkler viscoelastic model is proposed for the theoretical analysis of the indentation strain of rubber track road wheels. The model includes the following assumptions. (1) The springs of the rubber track along the direction of dent deformation are independent. Shear interactions between adjacent elements of the viscoelastic material are ignored. (2) The belt core is flexible and deforms with the substrate during the track depression process. Since the extrusion load is perpendicular to the length direction of the belt core, its influence on the bearing capacity of the rubber track is ignored. (3) The inertia of the rubber track material is negligible. (4) The road wheel is rigid, ignoring its own deformation.

As shown in Equation (8), in order to obtain the stress and strain of the contact section, the deformation of the contact section of the rubber track in the static state is firstly studied. The maximum indentation deformation $\delta \left(x\right)$ of the track and the strain $\varepsilon \left(x\right)$ at any point in the indentation area are:(8)$$\begin{array}{c} \left\{\begin{array}{c} \delta \left(x\right)=\sqrt{R^{2}-x^{2}}-\sqrt{R^{2}-a^{2}}\\ \varepsilon \left(x\right)=\frac{\delta \left(x\right)}{H}=\frac{\sqrt{R^{2}-x^{2}}-\sqrt{R^{2}-a^{2}}}{H} \end{array}\right.\\ s.t.-a<x<a \end{array}$$

Since the constitutive formula of the rubber track is: (9)$$\sigma \left(x\right)=E_{\mathrm{rubber}}\left(w,T\right)\varepsilon \left(x\right)$$
(10)$$F_{i}=\int_{-a}^{a}E_{\mathrm{rubber}}\left(w,T\right)\varepsilon \left(x\right)\mathrm{Ldx}$$
where $F_{i}$ is the i-th road wheel load; R is the radius of the road wheel; δ is the static maximum indentation depth; a is half the contact length; H is the track thickness; L is the axial length of the road wheel.

As shown in Figure 5, the constitutive model and load of the rubber track are taken as known conditions in the calculation and analysis process. The contact length is calculated cyclically and iteratively using a numerical calculation method. As shown in Equation (11), the length of the contact area is assumed to be a linear function of the radius. According to this function, the strain and stress of the contact area are calculated, and the cumulative reaction force $F_{i}\left(j\right)$ generated by the contact area is calculated integrally. The solution ends when the error between the reaction force $F_{i}\left(j\right)$ and the load $F_{i}$ is less than 1%. The values of contact area length, stress, and strain are calculated.
(11)$$a\left(j\right)=\frac{j\times R}{1000}$$

As shown in Figure 6, the changing trend of the contact length and strain amplitude of the load wheel with load is simulated. The initial conditions are that the diameter of the road wheel is 0.26 m, the elastic modulus of the rubber is 40 Mpa, the load range is 1–5 kN, and the interval is 1 kN. The analysis results show that the contact length and strain amplitude of the road wheel increase with the increase in load.

#### 2.4.2. Tension Strain Analysis

Excessive track tension will lead to increased energy consumption and reduced track life. Insufficient tension will increase the risk of rubber track decoupling. Reasonable tension should ensure that the track has a reasonable amount of overhang. The rubber track between the driving wheel and the tensioning wheel is the slackest area of the high-speed rubber track assembly, and it is also the area with the greatest risk of track detachment. Therefore, the overhang of the rubber track is analyzed in the stress state of the loose side to ensure that the track will not be detached. Considering the influence of the elastic modulus in the tension direction on the sag of the rubber track, the elastic catenary equation of the rubber track is established. As shown in Figure 7, the length direction of the rubber track is set as the x-axis, the direction perpendicular to the plane of the rubber track is the y-axis, and the force analysis is carried out for the micro-element section of the rubber track.

As shown in Equation (12), the force balance equation of the micro-element section of the rubber track is:(12)$$\left\{\begin{array}{c} (F_{0}+\mathrm{dF})\cos(\theta +\Delta \theta )=F_{0}\cos\theta =F\\ \left(F_{0}+\mathrm{dF}\right)\sin(\theta +\Delta \theta )=F_{0}\sin\theta +qgdl_{s} \end{array}\right.$$

Then, the elastic catenary equation of the rubber track is obtained:(13)$$\left\{\begin{array}{c} x=\frac{F}{\mathrm{qg}}\left[\ln|\sec u+\tan u|+\varepsilon \tan u\right]+C_{1}\\ y=\frac{F}{\mathrm{qg}}\left(\sec u+\frac{1}{2}\varepsilon {\left(\sec u\right)}^{2}\right)+C_{2} \end{array}\right.\;s.t.\;u\in \left[0,\arctan\left(\frac{{\mathrm{qga}}_{j}}{F}\right)\right]$$

After simplification, we obtain:(14)$$l_{\max}=\frac{{\mathrm{qga}}_{j}^{2}}{2\mathrm{EA}}\frac{F}{\mathrm{qg}}\left[\sqrt{1+{\left(\frac{{\mathrm{qga}}_{j}}{F}\right)}^{2}}-1\right]\frac{h_{d}}{2}$$

In order to prevent the rubber crawler from falling off the chain, set the maximum overhang of the crawler to not exceed half the height of the driving teeth. Solving the above formula, the tension of the rubber track is calculated as follows:(15)$$F=\frac{\mathrm{gq}\left(A^{2}E_{s}{}^{2}h_{d}{}^{2}-4A^{2}E_{s}{}^{2}l^{2}-2{\mathrm{AE}}_{s}{\mathrm{gh}}_{d}l^{2}q+g^{2}l^{4}q^{2}\right)}{4A^{2}E_{s}{}^{2}h_{d}-4{\mathrm{AE}}_{s}{\mathrm{gl}}^{2}q}$$
where θ is the angle between the tension of the rubber track and the x-axis; $a_{j}$ indicates the length of the track between the driving wheel and the tensioning wheel; F is the tension of the rubber track; ε is the tensile strain of the track; q is the weight per unit length of the rubber track; g is the acceleration due to gravity; A is the cross-sectional area of the track; $E_{S}$ is the tensile modulus of elasticity of the track; $l_{\max}$ is the maximum sag of the track; $h_{d}$ is the driving tooth height.
(16)$$F=E_{S}{A\varepsilon }_{\mathrm{tension}}=\left(E_{\mathrm{wire}}A_{\mathrm{wire}}+E_{\mathrm{rubber}}\left(w,T\right)A_{\mathrm{rubber}}\right)\varepsilon_{\mathrm{tension}}$$

The tensile strain generated by the overall rubber track under the tension force is:(17)$$\varepsilon_{\mathrm{tension}}=\frac{F}{\left(E_{\mathrm{wire}}A_{\mathrm{wire}}+E_{\mathrm{rubber}}\left(w,T\right)A_{\mathrm{rubber}}\right)}$$

The tensile stress generated by the rubber matrix under the action of the tension spring is:(18)$$\sigma_{\mathrm{tension}}=E_{\mathrm{rubber}}\left(w,T\right)\varepsilon_{\mathrm{tension}}$$

#### 2.4.3. Bending Strain Analysis

As shown in Figure 8, bending strain is the internal bending strain formed by the rubber track bending along the wheel body. The rigidity of the steel wire reinforcement layer built into the track is much greater than that of the rubber. It can be assumed that the length of the wire reinforcement does not change when the track is bent.

The deformation at any height h of the rubber track is:(19)$$\Delta l=\theta_{\mathrm{rad}}(R+h)-\theta_{\mathrm{rad}}(R+h_{0})$$
(20)$$\varepsilon \left(h\right)=\frac{\theta_{\mathrm{rad}}(R+h)-\theta_{\mathrm{rad}}(R+h_{0})}{\theta_{\mathrm{rad}}(R+h_{0})}=\frac{h-h_{0}}{R+h_{0}}\;s.t.0\leq h\leq H$$

Combined with the rubber constitutive model, the bending stress along the thickness direction can be expressed as:(21)$$\sigma \left(h\right)=E\left(w,T\right)\varepsilon \left(h\right)$$
where $\sigma \left(h\right)$ is the bending stress along the thickness direction of the track; $\varepsilon \left(h\right)$ is the bending strain along the thickness direction of the track; R is the radius of the road wheel; $\theta_{\mathrm{rad}}$ is the curvature of the track; $h_{0}$ is the distance between the built-in belt core and the inner surface of the track.

#### 2.4.4. Driving Strain Analysis

The drive mode of the high-speed rubber track assembly is mainly divided into forced drive and friction drive. As shown in Figure 9, there are drive teeth on the inner surface of the forced drive rubber track, and the power is transmitted by the engagement of the drive teeth and the drive wheel. The forced drive can provide a more reliable torque transmission than the friction drive, and it is also the research object of this section.

The driving stress is the tensile stress generated by the meshing of the driving pin of the driving wheel and the driving teeth of the rubber track. The driving force $F_{D-i}$ of the driving pin is:(22)$$F_{D-i}=\frac{M_{D}}{{\mathrm{nR}}_{D}}$$
where $M_{D}$ is the driving torque; $R_{D}$ is the pitch circle radius; n is the number of teeth that the drive pin and the track drive teeth mesh at the same time.

The tensile strain produced by the rubber track under the driving force is:(23)$$\varepsilon_{\mathrm{drive}-i}=\frac{\sum_{k=1}^{i}F_{D-i}}{\left(E_{\mathrm{wire}}A_{\mathrm{wire}}+E_{\mathrm{rubber}}\left(w,T\right)A_{\mathrm{rubber}}\right)}$$

Under the driving action, the tensile stress of the rubber matrix is:(24)$$\sigma_{\mathrm{drive}-i}=E_{\mathrm{rubber}}\left(w,T\right)\varepsilon_{\mathrm{drive}-i}$$

#### 2.4.5. Strain Load Spectrum of Rubber Track

As shown in Figure 10, based on the road wheel depression strain, track tension strain, bending strain, and driving strain, the strain distribution diagram of each position of the track ring is constructed. During the driving process, the rubber track reciprocates cyclically, and the strain load spectrum of the high-speed rubber track is established.

The rainflow counting method is widely used in fatigue life analysis. Since the stress–strain hysteresis loop of the rubber material is similar to the cyclic variation curve of the fatigue stress–strain of the material, the rainflow counting method is used to complete the cycle counting of the stress–strain load spectrum of the rubber track. The rainflow counting method rotates the time-varying strain history data curve by 90°, and the time coordinate axis is vertically downward. The data records are like a series of roofs, and the rainwater flows down the roofs to record the number of data cycles [30,31]. The calculation program of the strain amplitude rainflow method was compiled. The hysteresis loop statistical analysis of the rubber track strain load spectrum was carried out to form the strain rainflow matrix shown in Figure 11.
(25)$$S=\left[\varepsilon_{\mathrm{range}},\varepsilon_{\mathrm{mean}},S_{\mathrm{cycle}}\right]$$
where S is the strain rainflow matrix of the rubber track; $\varepsilon_{\mathrm{range}}$ is the strain amplitude; $\varepsilon_{\mathrm{mean}}$ is the mean value of strain; $S_{\mathrm{cycle}}$ is the number of load cycles.

### 2.5. Establish Temperature Rise Model of Rubber Track

The hysteresis loss of the rubber track under alternating loads is converted into heat energy. At the same time, there is heat loss in the rubber track during driving. The difference between the heat generated and the heat lost is the internal energy of the rubber track. Considering the track weight, specific heat capacity, and ambient temperature, the real-time temperature of the rubber track is obtained, as shown in Equation (26).
(26)$$T=\frac{Q_{\mathrm{gen}}-Q_{\mathrm{dis}}}{{\mathrm{mC}}_{P}}+T_{0}$$
where T is the real-time average temperature of the rubber track; $T_{0}$ is the ambient temperature; $C_{P}$ is the heat capacity of the rubber; m is the mass of the track.

#### 2.5.1. Thermal Analysis of Hysteresis Loss

When a dynamic load is applied, the dynamic stress and dynamic strain borne by the rubber material are shown in Equation (27).
(27)$$\left\{\begin{array}{c} \sigma \left(t\right)=\sigma_{a}\sin\omega t\\ \varepsilon \left(t\right)=\varepsilon_{a}\sin(\omega t-\delta ) \end{array}\right.$$

As shown in Equation (28), the heat generation per unit time is the product of the frequency of cyclic loading, the loss modulus, and the square of the strain amplitude. The heat generation per unit of time is the product of the frequency of cyclic loading, the loss modulus, and the square of the strain amplitude. At the same time, the loss modulus is affected by many factors, such as temperature and excitation frequency. The loss modulus further affects the single-cycle hysteresis temperature rise of the rubber track.
(28)$$Q=\int_{0}^{2\pi /w}f\sigma \left(t\right)\frac{d\varepsilon \left(t\right)}{\mathrm{dt}}\mathrm{dt}={\pi f\varepsilon }_{a}^{2}\;E\;\prime \prime \left(w,T\right)$$
where $\sigma_{a}$ is the stress amplitude; $\varepsilon_{a}$ is the strain amplitude; ω is the angular frequency; t is time; δ is the phase difference between dynamic stress and dynamic strain; f is the number of excitations per unit time; Q is the heat generation per unit volume per unit time.

As shown in Equation (29), based on the idea of numerical calculation, the calorific value of the rubber track is the cumulative function of the strain amplitude, the number of cycles, and the loss modulus. At the same time, the loss modulus varies with the temperature and travel speed of the rubber track.
(29)$$Q_{\mathrm{gen}}=\overset{n=\frac{L_{d}}{L_{R}}}{\underset{i=1}{\sum }}{\pi S}_{\mathrm{cycle}-i}\varepsilon_{\mathrm{rang}-i}^{2}E_{i}{}^{\prime \prime }\left(w,T\right)$$
where $Q_{\mathrm{gen}}$ is the cumulative heat generation of the track; $\varepsilon_{\mathrm{rang}-i}$ is the strain amplitude of the i-th crawler cycle statistics; $S_{\mathrm{cycle}-i}$ is the number of cycles corresponding to the i-th crawler cyclic strain amplitude; $E_{i}{}^{\prime \prime }\left(w,T\right)$ is the loss modulus of the i-th track cycle; n is the number of cycles; $L_{d}$ stands for driving distance; $L_{R}$ is the length of track loop.

#### 2.5.2. Analysis of Convective Heat Dissipation

As shown in Equation (30), the heat dissipation of the rubber track is a function of convective thermal conductivity, track surface area, average track temperature, and ambient temperature. During the driving process, the rubber track is regarded as a flat plate to force turbulent heat dissipation in infinite space. Referring to the tire convective heat transfer coefficient empirical formula, the convective heat transfer coefficient of the rubber track increases linearly with the driving speed [32].
(30)$$Q_{\mathrm{dis}}={\lambda A}_{\mathrm{track}}\left(T-T_{0}\right)$$
(31)λ=44.39+1.295v
where $Q_{\mathrm{dis}}$ is the heat dissipation of the track; $A_{\mathrm{track}}$ is the track surface area; λ is the convective heat transfer coefficient; T is the real-time temperature of the rubber track; $T_{0}$ is the ambient temperature; v is driving speed.

### 2.6. Establish Rolling Resistance Torque Model of High-Speed Rubber Track Assembly

As shown in Equation (32), during the constant speed running of a tracked vehicle, based on the principle of energy conservation, the total energy input of the driving wheels is also the total energy consumption of the track assembly. At the macro level, the hysteresis energy loss of the rubber track assembly is reflected in the form of rolling resistance. The rolling resistance torque of the high-speed rubber track assembly is expressed as the sum of the rolling resistance torque equivalent to the hysteresis loss of the rubber track, the friction resistance torque of the wheel train bearing, the inertial resistance torque, and the wind resistance torque.
(32)$$M_{\mathrm{rolling}}=M_{\mathrm{hysteresis}}+M_{\mathrm{bearing}}+M_{\mathrm{wind}}+M_{\mathrm{inertial}}$$
where $M_{\mathrm{rolling}}$ is the rolling resistance torque; $M_{\mathrm{hysteresis}}$ is the rolling resistance torque equivalent to hysteresis loss; $M_{\mathrm{bearing}}$ is bearing frictional torque; $M_{\mathrm{wind}}$ is wind resistance torque; $M_{\mathrm{inertial}}$ is inertial resistance torque.

#### 2.6.1. The Torque Equivalent to Hysteresis Loss

The hysteresis loss of the rubber track is the energy loss of the rubber track under the action of alternating loads such as driving, depression, and bending during driving. At the macro level, hysteresis losses are reflected in the form of rolling resistance torque.
(33)$$M_{\mathrm{hysteresis}}=\frac{\partial Q_{\mathrm{gen}}}{w\partial t}=\frac{\partial Q_{\mathrm{gen}}}{w\partial t}$$

#### 2.6.2. Bearing Frictional Torque

Rolling bearings are arranged inside the wheel trains, such as drive wheels, road wheels, and guide wheels. During the rolling process, the bearing has frictional resistance. The bearing friction coefficient depends on several factors, such as bearing type and lubrication method. Referring to the empirical formula of bearing friction and rolling resistance of metal tracked vehicles, the bearing friction torque is shown in Equation (34).
(34)$$M_{\mathrm{bearing}}=\frac{f_{\mathrm{bearing}}G}{R_{1}}$$
where $f_{\mathrm{bearing}}$ is the friction coefficient of the bearing, the value is 0.0075 [33]; $R_{1}$ is the pitch circle radius of the driving wheel; G is the weight of the vehicle.

#### 2.6.3. Wind Resistance Torque

During high-speed driving, the vehicle constantly hits the air in front and generates wind resistance. Wind resistance is related to many factors, such as vehicle speed, body shape, windward area of the vehicle body, and volume of the vacuum area behind the vehicle. According to automobile theory, the wind resistance of an automobile can be expressed as Equation (35).
(35)$$M_{\mathrm{wind}}=\frac{{\mathrm{CAv}}^{2}}{76140R_{1}}$$
where C is the air resistance coefficient, the value is 0.8 [33]; A is the windward area; v is driving speed.

#### 2.6.4. Inertial Resistance Torque

According to the principle of inertia, the vehicle will generate inertial resistance when moving at variable speeds. When the vehicle accelerates, the inertial resistance is opposite to the direction of travel. When the car decelerates, the stored kinetic energy tries to keep the original speed and slide forward, and the inertial resistance is the same as the driving direction. Inertial resistance torque can be expressed as Equation (36).
(36)$$M_{\mathrm{inertial}}=\frac{G\partial v}{R_{1}\partial t}$$

## 3. Research Objects and Methods

### 3.1. Experimental Platform and Test System

#### 3.1.1. XGD24D Electric Drive Chassis

XGD24D electric drive chassis is a two-axle 4 × 4 off-road electric transmission chassis that can realize wheel-shoe interchange/track switching. It is suitable for operation in plateaus, mountains, hills, and plains below 3000 m. The operating system adopts a modular design and has multiple functions such as lifting, digging, crushing, grabbing, shearing, cutting, breaking barriers, dragging, high-rise buildings, and petrochemical oil tank fires.

As shown in Figure 12, the XGD24D electric drive chassis adopts a hybrid drive scheme, which has 5 driving modes: pure electric, engine drive, hybrid drive, brake recovery, and parking charging. It is equipped with the MC11.36 engine and adopts a driving strategy in which the rear axle follows the main front axle. The front axle adopts pure electric drive, matched with a three-phase permanent magnet synchronous motor. The rear axle adopts hybrid electric transmission mode, matching the three-phase permanent magnet synchronous motor. The gearbox is matched between the axle and the electric motor, and the front and rear axles are matched with a 6-speed gearbox. In the crawler driving mode, the required torque is relatively large, and the engine alone cannot meet the torque requirement of the vehicle. At this time, the engine, the torque booster TM motor, and the TM motor jointly provide power for the vehicle to meet the driving torque requirement of the crawler-type vehicle with a maximum speed of ≥50 km/h.

As shown in Figure 13, the XGD24D electric drive chassis has tire driving mode, rubber track driving mode, and wheel-track interchange function. The high-speed rubber track assembly and tires adopt a common mounting interface and are fixed to the chassis rim. In the wheeled driving mode, the XGD24D electric transmission chassis is equipped with 4 tires and adopts the Ackermann steering system. In the rubber track driving mode, the XGD24D electric drive chassis is equipped with 4 high-speed rubber track assemblies and is also equipped with an auxiliary steering system. The steering mode still conforms to the Ackermann steering law.

#### 3.1.2. Prototype of High-Speed Rubber Track Assembly

As shown in Figure 14, the high-speed rubber track assembly includes a driving wheel, tensioning wheel, road wheel, guide wheel, tensioning spring, and tensioning arm. The driving wheel and the rubber track are driven by teeth and are located on the top of the high-speed rubber track assembly. The tensioning wheel is located at the front of the high-speed rubber track assembly, and the tensioning of the track is realized by the joint action of the tensioning spring and the tensioning arm.

#### 3.1.3. Experimental Data Acquisition System

As shown in Figure 15, the experimental data acquisition system is mainly composed of a notebook computer, data acquisition software, DW43 data acquisition instrument, hydraulic pressure sensor, GPS speedometer, angle sensor, engine, and motor built-in sensors, etc. The real-time data of the pressure sensor, angle sensor, and GPS speedometer are collected by the DW43 data acquisition instrument. Data information such as engine speed, load, and gear position of the gearbox are collected through the on-board controller, and basic data are provided for subsequent test data processing. The data acquisition system adopts mature and stable test instruments and sensors to ensure the effectiveness of data acquisition. The specific specifications and models are shown in Table 1.

### 3.2. Methods and Parameters

#### 3.2.1. Initial Parameters

The load and structure parameters of high-speed rubber track assembly are shown in Table 2. The density, thermal conductivity, heat capacity, and other parameters in Table 2 are obtained by test. In this paper, we ignore the influence of frequency, temperature, and load on density, thermal conductivity, and heat capacity.

The rubber matrix is composed of 30% styrene-butadiene rubber, 2% zinc oxide, 5% silicon dioxide, 2% sulfur, 17% carbon black, 1% stearic acid, 0.5% antioxidant BLE, and residual natural rubber. Styrene butadiene rubber and natural rubber are sourced from Aladdin Industrial Co., Ltd., Shanghai, China. Other reagents are sourced from Nanjing Xinyue Chemical Industrial Co., Ltd., Nanjing, China. The above percentages are mass percentages. The dynamic properties of rubber materials are tested by the Q800 dynamic thermomechanical analyzer of TA company. The test frequency is 10 Hz. The test temperature range is −90 °C–120 °C. The storage modulus, loss modulus, and loss tangent of the rubber material are obtained by testing, as shown in Figure 16. 

#### 3.2.2. Experimental Method

In order to verify the validity of the theoretical model of rolling resistance in the second chapter. The experimental test and simulation analysis were carried out for the working conditions of the maximum speed of 10 km/h, 30 km/h, and 50 km/h, respectively. The errors of simulated data and experimental data are compared.

Aiming at the working conditions of the maximum speed of 10 km/h and 30 km/h, the test was carried out at the circular test site in the comprehensive test site of a construction machinery enterprise. The road surface of the site is flat and solid, the site diameter is ≥100 m, and the road slope does not exceed 1%, which meets the test requirements. In order to simulate the dynamic change process, the vehicle was driven around the outer ring of the circular field for testing. The turning radius of the vehicle is about 100 m. The vehicle stops and saves the data every time it drives 1 lap and then accelerates to the maximum driving speed.

Due to the test of the maximum speed of 50 km/h, there is a certain danger in the circular test site. Therefore, the high-speed driving test with a maximum speed of 50 km/h was carried out on a straight road with a length of 2000 m in the proving ground. The road surface is flat and solid, and the road slope does not exceed 1%, which meets the test requirements.

During the test, the XGD24D electric drive chassis was tested at high speed in the rubber track driving mode. The vehicle accelerates continuously from a standstill to the maximum speed. Data information such as engine speed, torque, and vehicle speed is collected by the test data acquisition system. According to vehicle information such as engine speed, torque, transmission ratio, and axle ratio, the driving torque of the driving wheel is converted, which is also the rolling resistance torque of the high-speed rubber track assembly. During the test, the driver increased the speed of the vehicle to the set speed as soon as possible and kept it stable. In order to maintain the consistency of handling, all tests were completed by the same driver.

## 4. Results and Discussion

### 4.1. Comparison of Results

In the circular test site, the test with a maximum driving speed of 10 km/h was carried out. The driving speed test data of the XGD24D electric drive chassis can be described by Figure 17. The test data is also the input parameter for the simulation of the rolling resistance torque model.

As shown in Figure 18, the blue straight line represents the variation trend of the simulated rolling resistance torque, and the red dotted line represents the variation trend of the experimental rolling resistance torque. The simulation results show that when the maximum vehicle speed is 10 km/h, both the simulated rolling resistance torque and the test rolling resistance torque show a gradual decrease trend and gradually tend to a stable value. In the range of 0–2500 s, the average absolute error MAE of the simulated rolling resistance torque and the experimental torque is 409 Nm, and the coefficient of determination R-square is 0.85, which shows a high degree of agreement.

In the circular test site, the test with a maximum driving speed of 30 km/h was carried out. The driving speed test data of the XGD24D electric transmission chassis can be described by Figure 19, which is also the input parameter of the rolling resistance torque model simulation.

As shown in Figure 20, the simulation results show that the blue straight line represents the variation trend of the simulated rolling resistance torque at a maximum speed of 30 km/h, and the red dotted line represents the variation trend of the test torque at a maximum speed of 30 km/h. The simulation results show that when the maximum vehicle speed is 30 km/h, both the simulated rolling resistance torque and the test torque show a gradual decrease trend and gradually tend to a stable value. In the range of 0–600 s, the average absolute error MAE of the simulated rolling resistance torque and the experimental torque is 707 Nm, and the coefficient of determination R-square is 0.88, which shows a high degree of agreement.

On the straight-line driving test site, the test with a maximum driving speed of 50 km/h was carried out. The driving speed test data of the XGD24D electric transmission chassis can be described by Figure 21, and the driving speed test data are also the input parameter of the rolling resistance model simulation.

As shown in Figure 22, the simulation results show that the blue straight line represents the variation trend of the simulated rolling resistance torque at a maximum speed of 50 km/h, and the red dotted line represents the variation trend of the test torque at a maximum speed of 50 km/h. When the maximum vehicle speed is 50 km/h, both the simulated rolling resistance torque and the test torque show a gradual decrease trend. In the range of 0–155 s, the average absolute error MAE between the simulated rolling resistance torque and the experimental torque is 1061 Nm, and the coefficient of determination R-square is 0.91, which shows a high degree of agreement.

### 4.2. Trend Analysis of Rolling Resistance Torque

The rolling resistance torque of a high-speed rubber track assembly is affected by many factors, such as driving speed, load, and ambient temperature. In order to facilitate the design and optimization of high-speed rubber track assembly, the discussion and analysis are carried out for the factors such as driving speed, load, and ambient temperature.

#### 4.2.1. Influence of Speed on Rolling Resistance Torque

Speed is one of the main parameters of high-speed rubber track assembly, so it is necessary to discuss it in depth. As shown in Figure 23, when the simulated driving speed is 10 km/h, 20 km/h, 30 km/h, 40 km/h, and 50 km/h, the acceleration time is set to 50 s, and the speed remains stable after reaching the preset driving speed. The total duration of the simulation is about 2500 s. During the process, the variation trend of the rolling resistance torque of the high-speed rubber track assembly is simulated and analyzed.

In order to compare the impact of driving speed, the acceleration time is set to 50 s. Due to the different accelerations on different curves, the simulated data for the first 50 s cannot be used for analysis. The simulation data of rolling resistance torque of high-speed rubber track assembly during 50–2500 s are mainly analyzed. As shown in Figure 24, the simulation analysis results show that under continuous and stable driving conditions, the rolling resistance torques at different driving speeds gradually decrease with time and tend to the same stable value. Comparison of rolling resistance torque simulation curves at different driving speeds. When the driving speed is high, the temperature rise of the rubber track is faster, and the time for the rolling resistance torque to approach a stable value is shorter. When the driving speed is low, the temperature rise of the rubber track is slow, and the time for the rolling resistance torque to approach a stable value is longer. At the same time point, the lower the speed, the larger the rolling resistance torque, and the higher the speed, the smaller the rolling resistance torque, and the rolling resistance difference at different speeds decreases gradually.

#### 4.2.2. Influence of Load on the Rolling Resistance Torque

Assuming that the vehicle speed is accelerated from 0 km/h to 50 km/h within 50 s, and the speed remains stable after reaching the preset speed, the total duration of the simulation is 2500 s. The rolling resistance torque trend of the high-speed rubber track assembly with the weight of 200 kN, 250 kN, 300 kN, and 350 kN is analyzed.

In order to facilitate the comparison of the influence of driving speed, the acceleration time is set to 50 s, and the acceleration of different curves is different, so the simulation data of the first 50 s is not used as the analysis basis. The simulation data of rolling resistance torque of high-speed rubber track assembly during 50–2500 s are mainly analyzed. As shown in Figure 25, the simulation analysis results show that under the condition of constant speed driving, the rolling resistance torque of high-speed rubber track assemblies with different loads gradually decreases with time and tends to different stable values. At any moment, the rolling resistance moment will increase as the vehicle load increases.

#### 4.2.3. Influence of Ambient Temperature on Rolling Resistance Torque

Because the accumulation of energy in the acceleration section will interfere with the law of the influence of ambient temperature on the rolling resistance torque of the high-speed rubber track assembly. Assume that the vehicle travels at a constant speed of 50 km/h and the vehicle weighs 300 kN. The total duration of the simulation is 1000 s. The variation trend of rolling resistance torque was simulated at −15 °C, 0 °C, 15 °C, and 30 °C, respectively.

As shown in Figure 26, the simulation analysis results show that under different ambient temperature, the rolling resistance moment of the high-speed rubber track assembly gradually decreases and tends to the same stable value. Within 0–250 s, the rolling resistance torque of the low ambient temperature is higher than that of the high ambient temperature. However, after 250 s, the rolling resistance torque at high ambient temperatures is higher. The reason for this phenomenon is that the rubber loss modulus is relatively large at low temperature, and the rolling resistance moment is relatively large at the initial stage of 0–250 s. At the same time, a large amount of heat is also generated, and the heat dissipation capacity of the rubber track is weak; the temperature of the track rises rapidly, and the loss modulus of the rubber decreases. After 250 s, the rolling resistance torque at low ambient temperature is slightly higher than that at high ambient temperature, but with the accumulation of time, the rolling resistance torque will tend to a stable value.

### 4.3. Promotion and Application

The theoretical model studied in this paper is applicable to the rolling resistance analysis of any rubber track assembly. The structural dimensions, load parameters of the rubber track assembly, and the dynamic mechanical performance parameters of the rubber composite track are known. First, the constitutive model of the rubber track is established. Then, the strain load analysis of the rubber track is carried out, and the strain load matrix is established by the rain flow method. The heat value of the rubber track is calculated by loss modulus and strain load matrix. Combined with the ambient temperature, the heating capacity of the rubber track is calculated, and the real-time temperature of the rubber track is obtained. According to the real-time temperature and constitutive model of the rubber track, the real-time loss modulus of the rubber track is obtained. The rolling resistance of the rubber track assembly is calculated, combined with the loss modulus and strain load matrix.

## 5. Conclusions

In this paper, based on the structure and load of the high-speed rubber track assembly, the indentation strain, bending strain, driving strain, and tension strain of the rubber track were analyzed, and the strain load spectrum was established. The strain load amplitude was extracted by the rainflow method, and the strain rainflow matrix was established. Combined with the constitutive model of rubber material, the trend of temperature rise and hysteresis loss of rubber track was given. The hysteresis loss of the rubber track is equivalent to the rolling energy loss of the high-speed rubber track assembly. Combining bearing frictional resistance torque, wind resistance torque, and inertial resistance torque, the rolling resistance torque model of high-speed rubber track assembly was established, and the following conclusions were drawn:The rolling resistance torque of the high-speed rubber track assembly is affected by the dynamic mechanical properties of the rubber track, showing the characteristics of thermal-mechanical coupling. In the initial state, the rolling resistance torque of the high-speed rubber track assembly is relatively large. Due to the hysteresis heat generation characteristics of rubber materials, the track temperature gradually increases with the increase in driving time. As a result, the loss modulus of the rubber is reduced, and the rolling resistance torque of the high-speed rubber track assembly is reduced.The rolling resistance torques at different driving speeds gradually decrease with time and tend to the same stable value. The time for the rolling resistance torque to decrease to a stable value decreases with the increase driving speed. Otherwise, it takes longer. At the same time point, the rolling resistance torque of the high-speed rubber track assembly decreases with the increase speed.The rolling resistance torque of high-speed rubber track assemblies with different weights gradually decreases with time and tends to different stable values. At any moment, the rolling resistance moment will increase as the vehicle load increases.Under different ambient temperatures, the rolling resistance torque decreases gradually and tends to the same stable value. Within 0–250 s, the rolling resistance torque of the low ambient temperature is higher than that of the high ambient temperature, but after 250 s, the rolling resistance torque of the high ambient temperature is slightly higher than that of the low ambient temperature. The reason is that the rubber loss modulus is relatively large at low temperature, and the rolling resistance torque is relatively large at the initial stage of 0–250 s. At the same time, a large amount of heat is generated, and the heat dissipation capacity is weak, the temperature of the track rises rapidly, and the loss modulus of the rubber decreases. After 250 s, the rolling resistance torque at low ambient temperature is slightly higher than that at high ambient temperature, but with the accumulation of time, the rolling resistance torque will tend to a stable value.

## Figures and Tables

**Figure 1 polymers-15-01642-f001:**
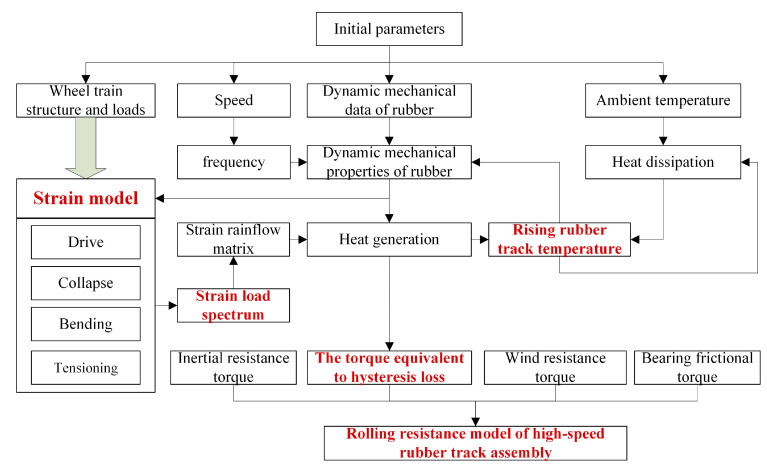
Research method of rolling resistance of high-speed rubber track assembly.

**Figure 2 polymers-15-01642-f002:**
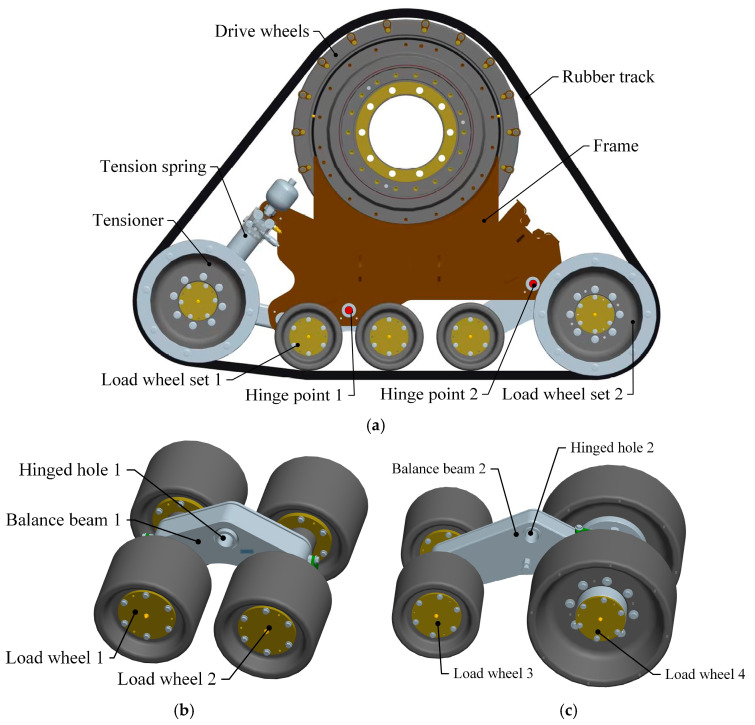
Components of high-speed rubber track assembly: (**a**) High-speed rubber track assembly; (**b**) Load wheel set 1; (**c**) Load wheel set 2.

**Figure 3 polymers-15-01642-f003:**
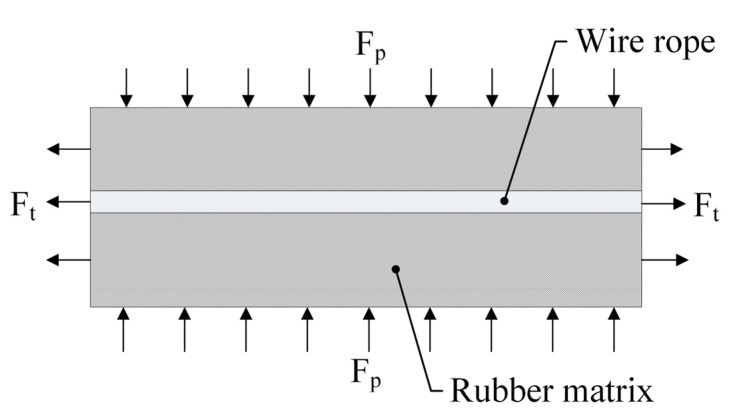
Schematic diagram of rubber track composition.

**Figure 4 polymers-15-01642-f004:**
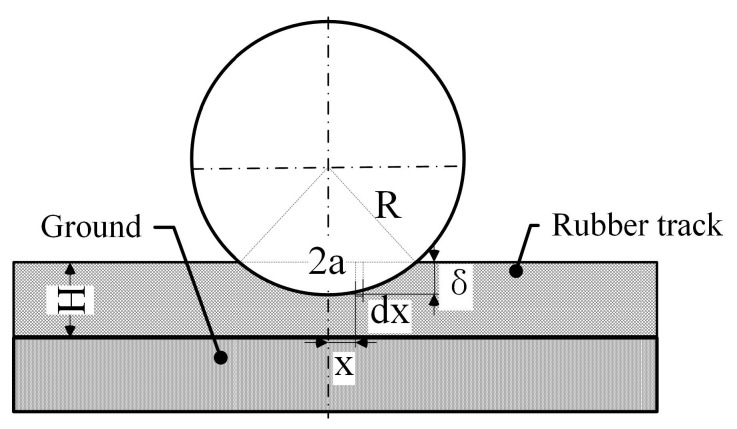
Stress and strain analysis of the contact area.

**Figure 5 polymers-15-01642-f005:**
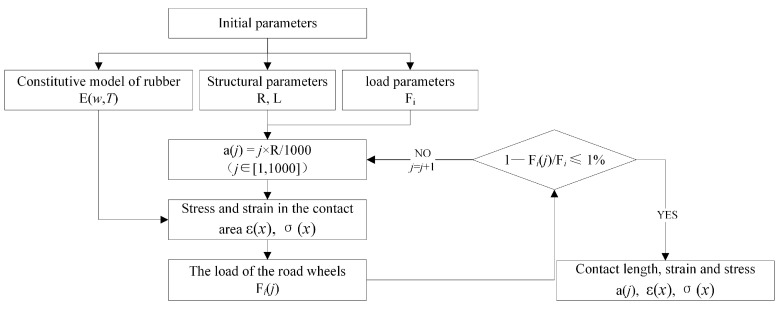
Calculation algorithm of contact length and stress and strain.

**Figure 6 polymers-15-01642-f006:**
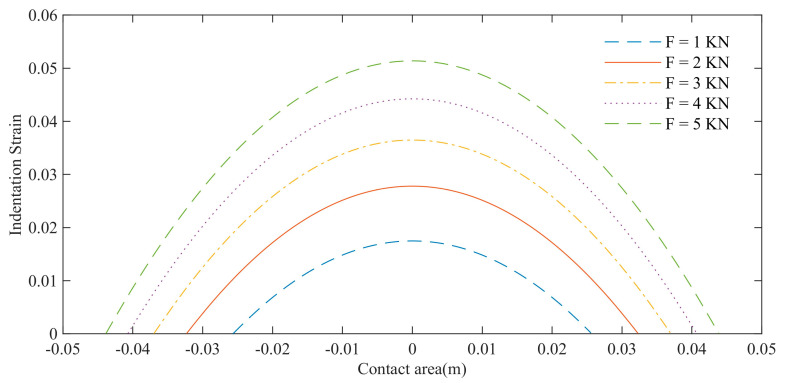
Influence of load on contact length and strain of load bearing wheel.

**Figure 7 polymers-15-01642-f007:**
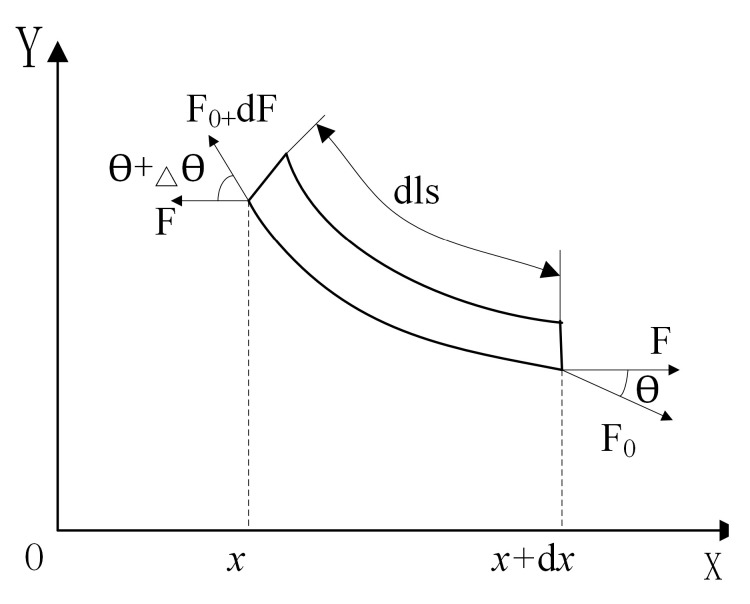
The force balance state of rubber track suspension element.

**Figure 8 polymers-15-01642-f008:**
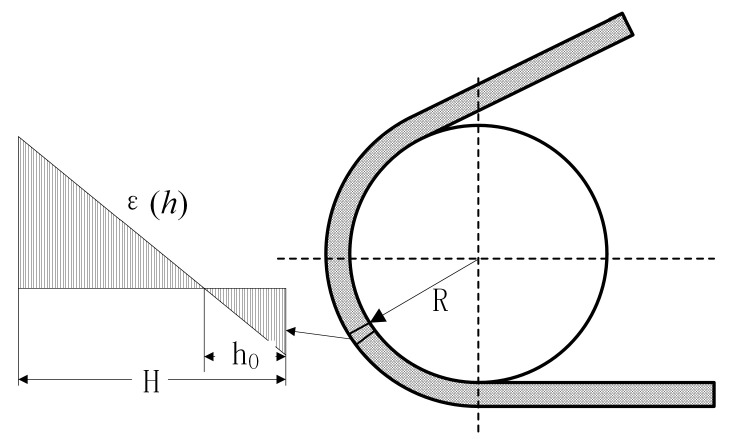
Bending stress–strain state of rubber track.

**Figure 9 polymers-15-01642-f009:**
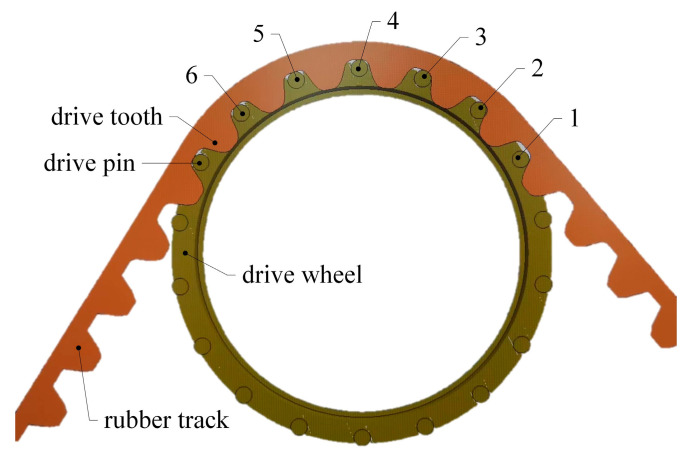
Schematic diagram of engagement between drive pin and drive tooth.

**Figure 10 polymers-15-01642-f010:**
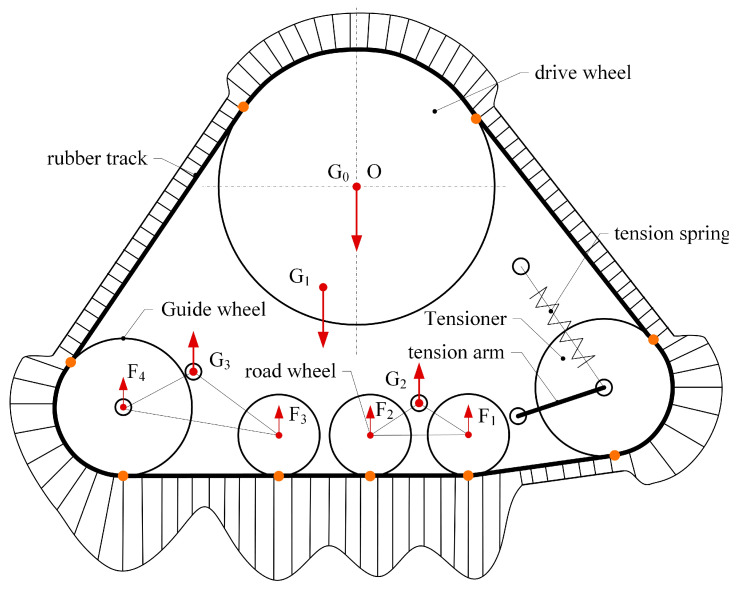
Strain load spectrum of rubber track.

**Figure 11 polymers-15-01642-f011:**
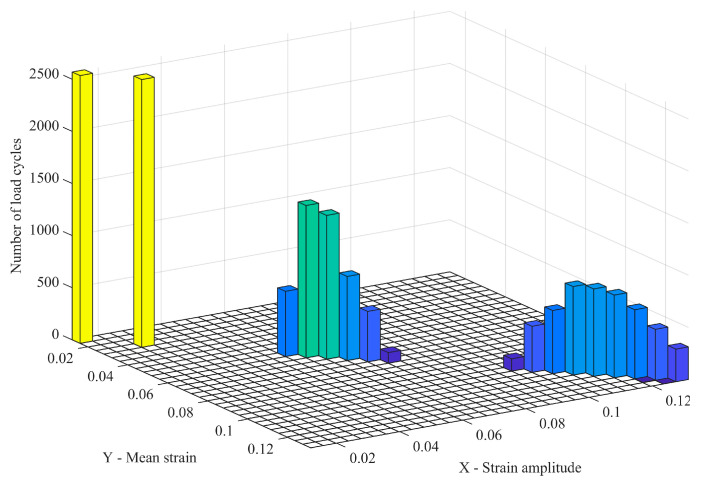
Strain Rainflow Matrix for Rubber Tracks.

**Figure 12 polymers-15-01642-f012:**
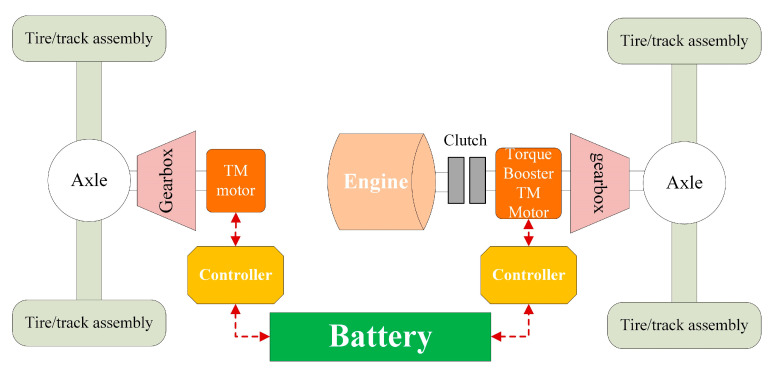
Driving Scheme of XGD24D Electric Drive Chassis.

**Figure 13 polymers-15-01642-f013:**
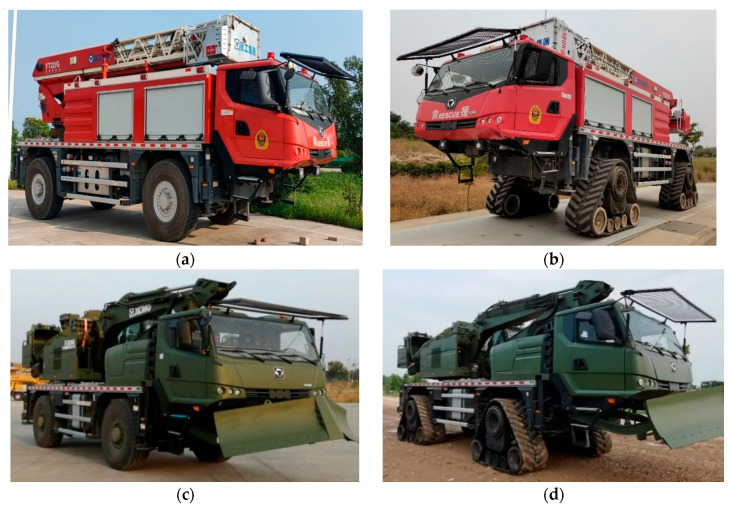
XGD24D electric drive chassis with tire and rubber track driving mode: (**a**) Tire driving mode and aerial ladder fire rescue operation system; (**b**) Rubber track driving mode and aerial ladder fire rescue operation system; (**c**) Tire driving mode and multi-functional emergency rescue operation system; (**d**) Track driving mode and multi-functional emergency rescue operation system.

**Figure 14 polymers-15-01642-f014:**
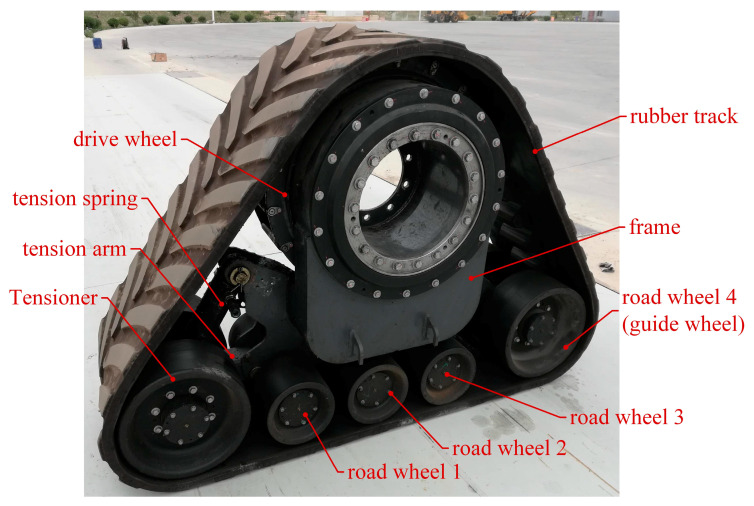
Physical prototype of high-speed rubber track assembly.

**Figure 15 polymers-15-01642-f015:**
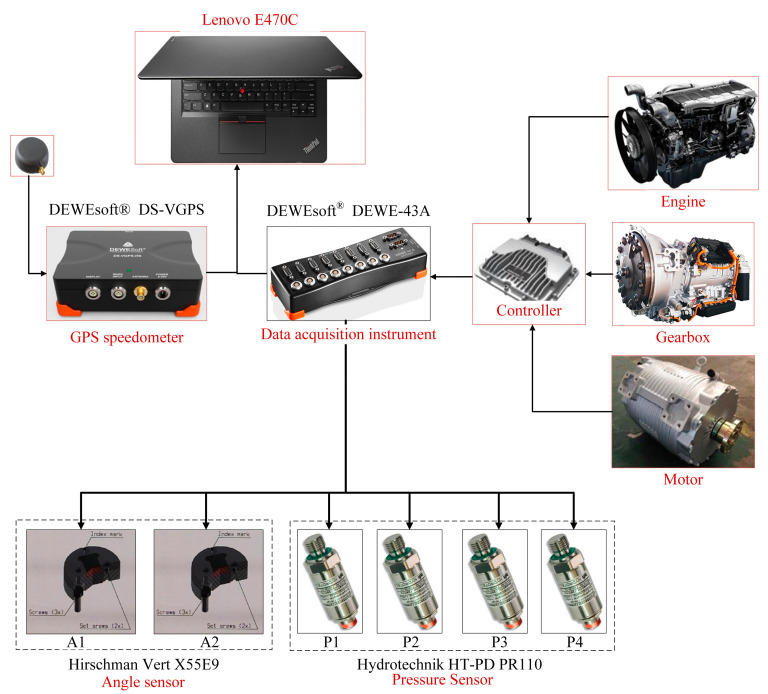
Experimental data acquisition system.

**Figure 16 polymers-15-01642-f016:**
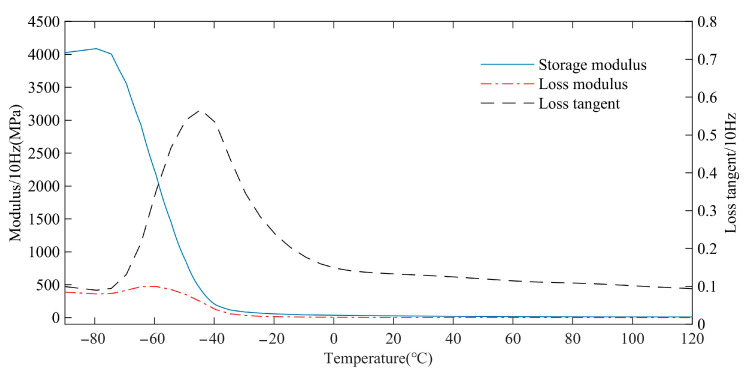
Dynamic parameters of rubber materials.

**Figure 17 polymers-15-01642-f017:**
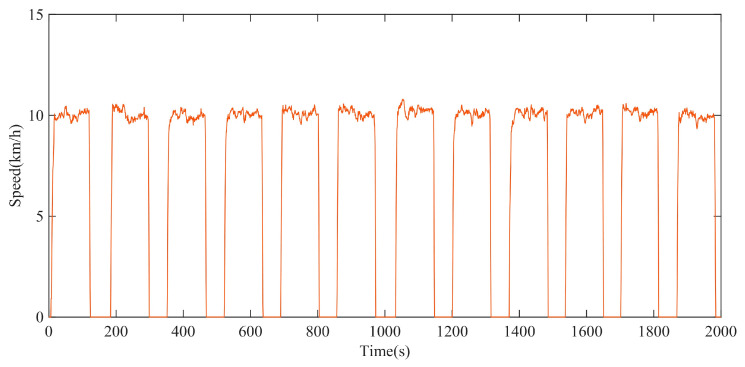
Test data of driving speed with maximum speed of 10 km/h.

**Figure 18 polymers-15-01642-f018:**
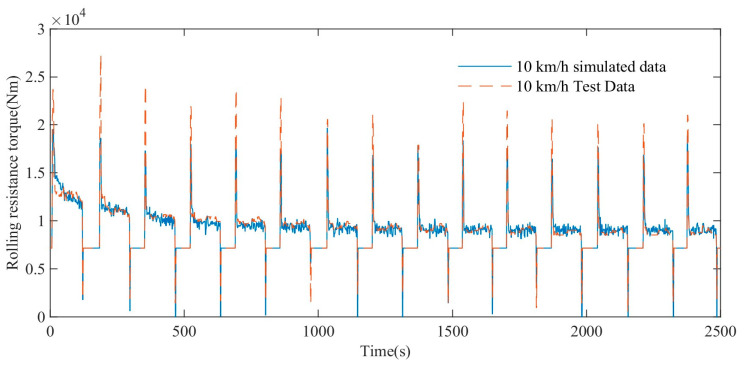
Comparison of rolling resistance torque test and simulation data at maximum speed of 10 km/h.

**Figure 19 polymers-15-01642-f019:**
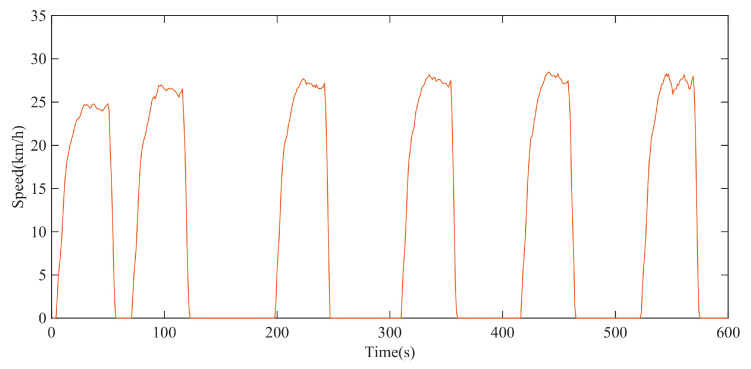
Test data of driving speed with maximum speed of 30 km/h.

**Figure 20 polymers-15-01642-f020:**
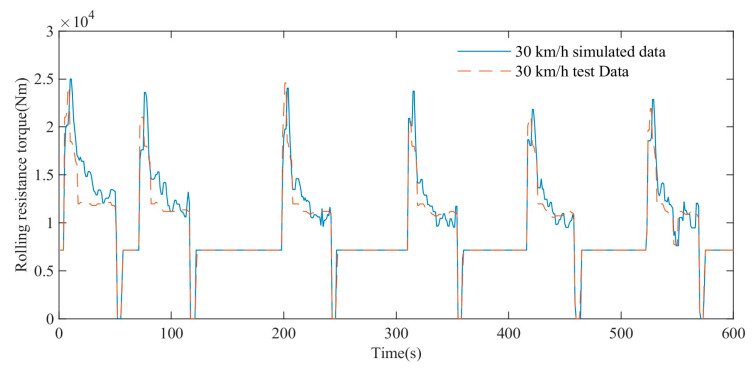
Comparison of rolling resistance torque test and simulation data at maximum speed of 30 km/h.

**Figure 21 polymers-15-01642-f021:**
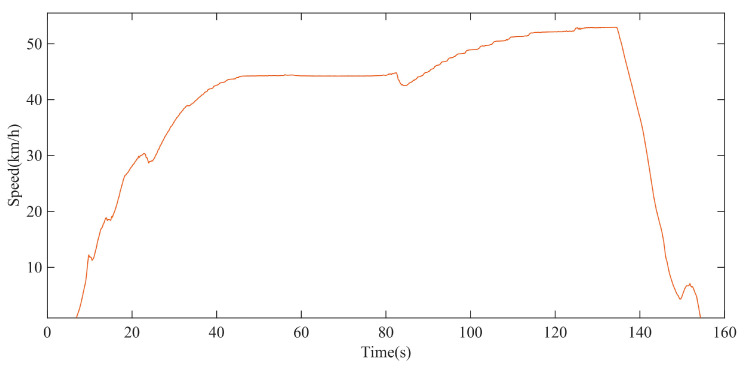
Test data of driving speed with maximum speed of 50 km/h.

**Figure 22 polymers-15-01642-f022:**
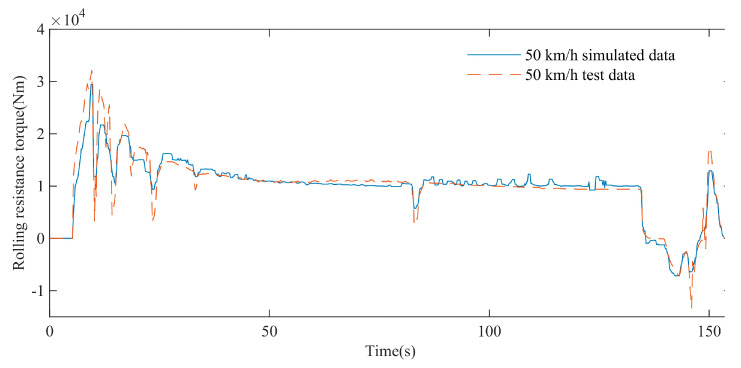
Comparison of rolling resistance torque test and simulation data at maximum speed of 50 km/h.

**Figure 23 polymers-15-01642-f023:**
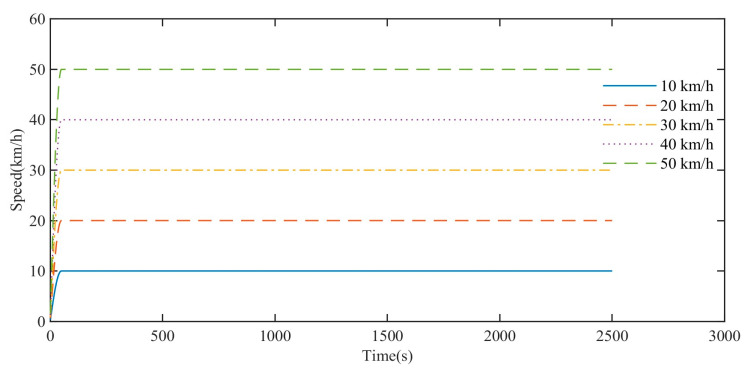
Simulated speed data.

**Figure 24 polymers-15-01642-f024:**
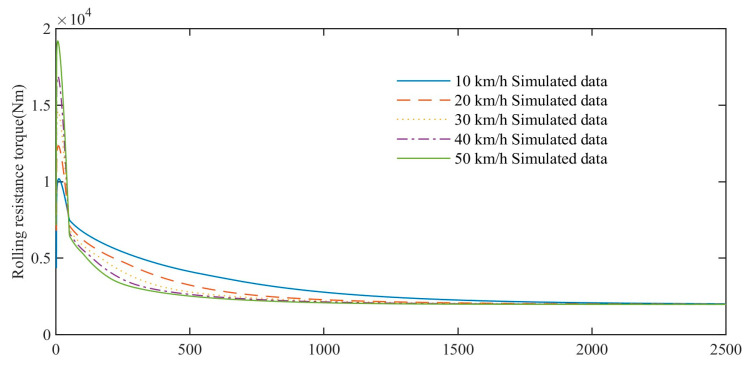
Effect of speed on rolling resistance torque.

**Figure 25 polymers-15-01642-f025:**
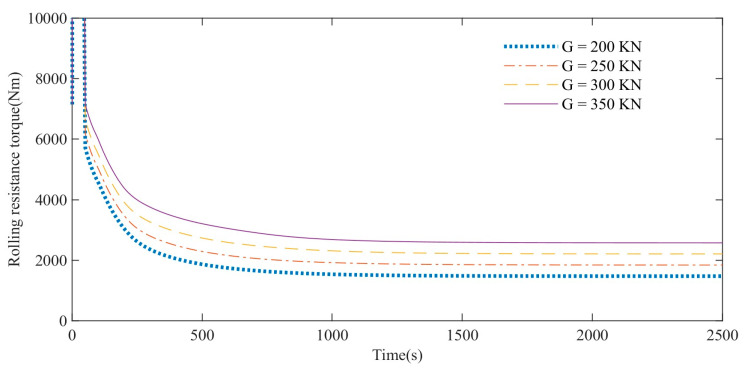
Influence of load on rolling resistance.

**Figure 26 polymers-15-01642-f026:**
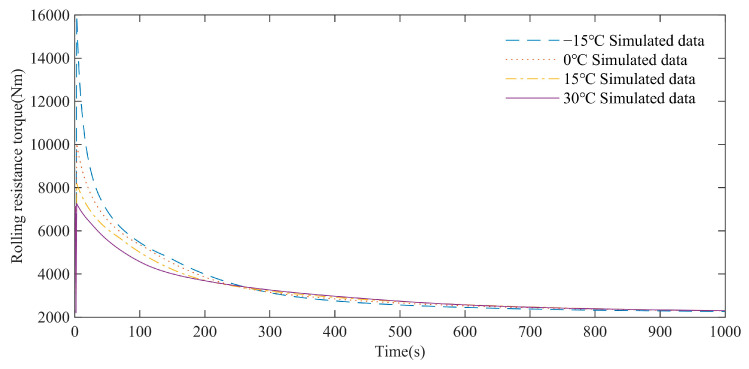
Influence of ambient temperature on rolling resistance.

**Table 1 polymers-15-01642-t001:** The experimental data acquisition system parameters.

Equipment	Manufacturer/Model	Quantity	Range	Frequency	Resolution
Laptop computer	Lenovo E470C (Lenovo Holdings Co., Ltd., Beijing, China)	1	-	-	-
Data acquisition instrument	DEWEsoft^®^ DEWE-43A (DEWEsoft Test Equipment Co., Ltd., Beijing, China)	1	-	-	-
Data acquisition software	DEWEsoft^®^ X3 SP10(DEWEsoft Test Equipment Co., Ltd., Beijing, China)	1	-	100 Hz	-
Hydraulic pressure sensor	Hydrotechnik HT-PD PR110 (Shanghai Zhong Ye Measurement&Technology Co., Ltd., Shanghai, China)	4	0–600 bar	100 Hz	0.01 bar
Rotation angle sensor	Hirschman Vert X55E9(Hersman Electronics (Shanghai) Co., Ltd., Shanghai, China)	2	−30°–30°	100 Hz	0.01°

**Table 2 polymers-15-01642-t002:** Parameters of theoretical analysis.

Description	Notation	Units	Values
Axle load	$G_{0}$	kN	112
Weight of track group	$G_{1}$	kN	12
Number of roller wheels	n	-	4
The contact area of the tread	A	m^2^	0.204
The thickness of the track pattern	d	m	0.005
Width of rubber track	W	m	0.635
Rubber mass density	ρ	kg/m^3^	1000
The rubber heat conductivity	λ	W/m/K	0.26
Heat capacity of rubber	$C_{P}$	J/kg/K	1500

## Data Availability

The data that support the findings of this study are available from the corresponding author upon reasonable request.

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
