# Peer review of "Modeling and Verification of Rolling Resistance Torque of High-Speed Rubber Track Assembly Considering Hysteresis Loss"

_polymers, 2023, doi:10.3390/polym15071642_

Round 1

Reviewer 1 Report

As for the temperature dependence of the rolling resistance.

 The authors show the temperature dependence of the rolling resistance in Figure 26.  I would like to know if time-temperature superposition principle is valid according to the WLF equation for the temperature dependence of the relaxation on the rolling resistance. If this is the case, it would eliminate the need to investigate at various temperatures.

As for Figures

Figure 21 : The units for the both axes of the graph are not indicated.

Figure 24 : The units for the horizontal axis of the graph are not indicated.

Author Response

Thank you for your letter and for the reviewers’ comments concerning our manuscript entitled “Modeling and Verification of Rolling Resistance Torque of High-speed Rubber Track Assembly Considering Hysteresis Loss” (Manuscript ID: polymers-2277759). Those comments are all valuable and very helpful for revising and improving our paper, as well as the important guiding significance to our researches. We have studied the comments carefully and tried our best to improve our manuscript. The point to point response to the reviewer’s comments are listed as following.

Q1: The authors show the temperature dependence of the rolling resistance in Figure 26.  I would like to know if time-temperature superposition principle is valid according to the WLF equation for the temperature dependence of the relaxation on the rolling resistance. If this is the case, it would eliminate the need to investigate at various temperatures.

Response:

The influence of ambient temperature on the rolling resistance of rubber track assembly is studied, as shown in Figure 26. The ambient temperature not only determines the initial temperature of the rubber track, but also affects the heat dissipation of the rubber track, thus affecting the temperature rise rate of the rubber track during driving, as discussed in Section 2.5.2. However, according to the WLF equation and the principle of temperature-frequency equivalence, it can only describe the effect of the temperature and excitation frequency of the rubber track on the rolling resistance, but not the effect of heat dissipation on the rolling resistance. Therefore, it is necessary to discuss the impact of ambient temperature on rolling resistance and retain the relevant discussion in Figure 26.

Q2: Figure 21 : The units for the both axes of the graph are not indicated.

Response:

The title and unit of the two coordinate axes of the figure have been added, as shown in Figure 21.

Q3: Figure 24 : The units for the horizontal axis of the graph are not indicated.

Response:

The units of the two coordinate axes of the figure have been added, as shown in Figure 24.

Finally, we appreciate very much for your time in editing our manuscript and the referees for their valuable suggestions and comments. I am looking forward to hearing from your final decision when it is made.

Best regards.

Reviewer 2 Report

In the presented manuscript entitled “Modeling and Verification of Rolling Resistance Torque of High-speed Rubber Track Assembly Considering Hysteresis Loss” (polymers-2277759), the authors investigated the rolling resistance of high-speed rubber assembly. They established and tested the new model. The rolling resistance of tires is well researched and covered in literature, but any new research in this field is still of interest. The novelty of this paper is development of new modeling approach for the specific rubber track assembly. The research is interesting and comprehensive. However, this paper contains several issues:

1.  Please emphasize the scientific contribution and novelty of presented approach.

2. The authors presented the research conducted on the specific construction of rubber track assembly. Please add section describing how this approach can be utilized for other systems.

3.  Authors should provide more information about rubber composition and type.

4. In the table 2 authors presented the rubber properties (density, heat conductivity, heat capacity). How were these properties determined or evaluated? Are these parameters independent from other process parameters (temperature, speed, load etc.)?

5.     Please update references with current research in this field.

Author Response

Thank you for your letter and for the reviewers’ comments concerning our manuscript entitled “Modeling and Verification of Rolling Resistance Torque of High-speed Rubber Track Assembly Considering Hysteresis Loss” (Manuscript ID: polymers-2277759). Those comments are all valuable and very helpful for revising and improving our paper, as well as the important guiding significance to our researches. We have studied the comments carefully and tried our best to improve our manuscript. The point to point response to the reviewer’s comments are listed as following.

Q1: Please emphasize the scientific contribution and novelty of presented approach.

Response:

A new theoretical model of the rolling resistance of the rubber track assembly considering the hydrogen energy loss is proposed In this model, the lag energy loss of the rubber track is innovatively equivalent to the energy consumption of the rolling resistance of the track assembly, providing a new method for the study of the rolling resistance of the high-speed rubber track assembly.

The above scientific contributions and novelty are supplemented in this article between lines 122 and 126.

Q2: The authors presented the research conducted on the specific construction of rubber track assembly. Please add section describing how this approach can be utilized for other systems.

Response:

The theoretical model studied in this paper is applicable to the rolling resistance analysis of any rubber track assembly. The structural dimensions, load parameters of the rubber track assembly and the dynamic mechanical performance parameters of the rubber composite track are known. First, the constitutive model of rubber track is established. Then, the strain load analysis of the rubber track is carried out, and the strain load matrix is established by the rain flow method. The heat value of rubber track is calculated by loss modulus and strain load matrix. Combined with the ambient temperature, the heating capacity of the rubber track is calculated, and the real-time temperature of the rubber track is obtained. According to the real-time temperature and constitutive model of rubber track, the real-time loss modulus of rubber track is obtained. The rolling resistance of the rubber track assembly is calculated, combined with the loss modulus and strain load matrix.

The above discussion has been added to Section 4.3 of the article.

Q3: Authors should provide more information about rubber composition and type.

Response:

The rubber matrix is composed of 30% styrene-butadiene rubber, 2% zinc oxide, 5% silicon dioxide, 2% sulfur, 17% carbon black, 1% stearic acid, 0.5% antioxidant BLE and residual natural rubber. The above percentages are mass percentages.

This part of the discussion has been supplemented between lines 513 and 515.

Q4: In the table 2 authors presented the rubber properties (density, heat conductivity, heat capacity). How were these properties determined or evaluated? Are these parameters independent from other process parameters (temperature, speed, load etc.)?

Response:

The density, thermal conductivity, heat capacity and other parameters in Table 2 are obtained by test. In this paper, we ignore the influence of frequency, temperature and load on density, thermal conductivity and heat capacity.

This part of the discussion has been supplemented between lines 509 and 511.

Q5: Please update references with current research in this field.

Response:

In this paper, the references numbered 2-6 and 17-24 have been updated to the latest.

Finally, we appreciate very much for your time in editing our manuscript and the referees for their valuable suggestions and comments. I am looking forward to hearing from your final decision when it is made.

Best regards
